# A New Genus of the *Microascaceae* (Ascomycota) Family from a Hypersaline Lagoon in Spain and the Delimitation of the Genus *Wardomyces*

**DOI:** 10.3390/jof10040236

**Published:** 2024-03-22

**Authors:** María Barnés-Guirado, Alberto Miguel Stchigel, José Francisco Cano-Lira

**Affiliations:** Mycology Unit, Medical School, Universitat Rovira i Virgili, C/Sant Llorenç 21, 43201 Reus, Spain; maria.barnes@urv.cat (M.B.-G.); jose.cano@urv.cat (J.F.C.-L.)

**Keywords:** endorheic, extremophiles, fungi, halophilic, halotolerant, phylogeny, Sordariomycetes, taxonomy

## Abstract

The Saladas de Sástago-Bujaraloz is an endorheic and arheic complex of lagoons located in the Ebro Basin and protected by the Ramsar Convention on Wetlands. Due to the semi-arid climate of the region and the high salinity of their waters, these lagoons constitute an extreme environment. We surveyed the biodiversity of salt-tolerant and halophilic fungi residents of the Laguna de Pito, a lagoon belonging to this complex. Therefore, we collected several samples of water, sediments, and soil of the periphery. Throughout the study, we isolated 21 fungal species, including a strain morphologically related to the family *Microascaceae*. However, this strain did not morphologically match any of genera within this family. After an in-depth morphological characterization and phylogenetic analysis using a concatenated sequence dataset of four phylogenetically informative molecular markers (the internal transcribed spacer region (ITS) of the nuclear ribosomal DNA (nrDNA); the D1-D2 domains of the 28S gene of the nuclear ribosomal RNA (LSU); and a fragment of the translation elongation factor 1-alpha (*EF-1α*) and the β-tubulin (*tub*2) genes), we established the new genus *Dactyliodendromyces*, with *Dactyliodendromyces holomorphus* as its species. Additionally, as a result of our taxonomic study, we reclassified the paraphyletic genus *Wardomyces* into three different genera: *Wardomyces sensu stricto*, *Parawardomyces* gen. nov., and *Pseudowardomyces* gen. nov., with *Parawardomyces ovalis* (formerly *Wardomyces ovalis*) and *Pseudowardomyces humicola* (formerly *Wardomyces humicola*) as the type species of their respective genera. Furthermore, we propose new combinations, including *Parawardomyces giganteus* (formerly *Wardomyces giganteus*) and *Pseudowardomyces pulvinatus* (formerly *Wardomyces pulvinatus*).

## 1. Introduction

Extreme environments are habitats where physical and/or chemical conditions are exceptionally hostile to the survival and proliferation of most life forms [1]. Typically, these harsh conditions are determined by remarkably high or low (“extreme”) temperatures or pH, a high salt concentration, osmolarity, hydrostatic pressure, UV radiation, low water activity, or a high concentration of toxic compounds, such as organic solvents and heavy metals [1,2]. Organisms that can survive and proliferate in such environmental conditions are known as “extremophiles”, and their ability to thrive relies on their distinctive biochemical machinery and physiology [3] and on their cell ultrastructure and composition [4].

Endorheic basins and their associated lagoons and lakes are landlocked hydrogeological structures that do not drain into large waterbodies (rivers, oceans), only experiencing water loss through evapotranspiration and percolation to the underground [5]. Commonly found in arid and semi-arid continental regions, these lagoons and lakes tend to be saline due to the gradual salt accumulation resulting from thousands of years of evaporative processes [5]. Though some of the world’s biggest lakes are endorheic (i.e., Great Salt Lake in North America, the Caspian and Aral Seas in Central Asia, and Lake Titicaca in South America), most of these sorts of lagoons and lakes are temporary, alternating dry and wet phases, as the evaporative process exceeds water inputs, mostly represented by precipitations [5,6]. Species inhabiting these lagoons are adapted to the arid/semi-arid climate, periodical droughts, and high salt concentrations, thus making them extremophiles [6,7].

The Saladas de Sástago-Bujaraloz is an endorheic and arheic (with no circulation of superficial water) complex of lagoons located at the center of the Ebro Basin, ~70 km southeast of Zaragoza city (Aragon Community) in the northeast of Spain [8,9]. The lagoon complex comprises more than one hundred basins positioned over a platform that is around 400 m.a.s.l. and more than 100 m above the Ebro River. The local climate is semi-arid, characterized by low rainfall rates and high evaporation rates that contribute to the biphasic wet/dry states and salt accumulation [9,10]. Due to its unique characteristics, 26 of these lagoons, representing the best-conserved and most representative, are protected by the Ramsar Convention on Wetlands [11], thus being part of the 10% of Spanish Ramsar sites that are inland saline wetlands [9]. Although some studies focused on the flora and the fauna, and the bacterial and archaeal microbiota [9,12,13,14] were conducted at the Saladas de Sástago-Bujaraloz, no fungal studies have been conducted yet.

The most commonly isolated fungi from hypersaline lagoons and lakes belong to the families *Aspergillaceae*, *Cladosporiaceae*, *Hypocreaceae*, *Pleosporaceae*, *Saccharomycetaceae*, and *Teratosphaeriaceae*, with members of the *Microascaceae* family being less frequently recovered [15,16,17,18]. The *Microascaceae* family was established by Luttrell (1951) to accommodate the genus *Microascus* [19]. Morphologically, members of the *Microascaceae* have asexual states predominantly characterized by the production of annellidic conidiogenous cells, forming unicellular or (more rarely) bicellular conidia, and sexual states producing closed or perithecial ascomata within soon evanescent asci and triangular, reniform, or lunate ascospores with or without germ pores [20,21]. This family includes fungi isolated from soil, decaying plant material, and air, and several species are pathogens for animals, including mammals and humans [20]. The *Microascaceae* currently comprises 23 genera, including *Acaulium*, *Brachyconidiellopsis*, *Cephalotrichum*, *Enterocarpus*, *Fairmania*, *Gamsia*, *Kernia*, *Lomentospora*, *Lophotrichus*, *Microascus*, *Parascedosporium*, *Petriella*, *Pseudallescheria*, *Pseudoscopulariopsis*, *Pithoascus*, *Polycytella*, *Rhexographium*, *Rhinocladium*, *Scedosporium*, *Scopulariopsis*, *Wardomyces*, *Wardomycopsis*, and *Yunnania*, and about 300 species [20,21,22,23,24]. The taxonomic position of the genus *Canariomyces* is controversial, as the phylogenetic analysis conducted by Wang et al. [25] correctly placed the genus in the *Chaetomiaceae* family, but Wang et al. [22] retain the genus in the *Microascaceae* family based on another molecular study. Among them, some species belonging to the genera *Lomentospora*, *Scedosporium*, and *Scopulariopsis* are frequently involved as pathogens in opportunistic infections in humans [26,27,28]. Since the reorganization of the family structure by Sandoval-Denis et al. [20], no further taxonomic adjustments have been made, except for Su et al. [29], who revised the genera *Acaulium* and *Kernia*. In addition, several new species belonging to this family have been described in more recent works [22,30,31,32,33].

During a survey on the fungal diversity of soils, lake sediments, and hypersaline waters carried out at the Laguna de Pito (one of the lagoons of Saladas de Sástago-Bujaraloz), we isolated several fungal taxa, including a strain showing morphological features of the *Microascaceae* family but not matching any previously described genera.

The main aim of this study was to show the fungal diversity inhabiting the Laguna de Pito, as well as to characterize phenotypically and to determine the phylogenetic placement of such fungal strains and other morphologically related taxa in the *Microascaceae*.

## 2. Materials and Methods

### 2.1. Sampling and Fungal Isolation

We collected several samples of water, sediments, and soil from the surrounding areas of Laguna de Pito in January 2022. This lagoon covers approximately 50 ha., dries intermittently, and is surrounded by fields designated for the cultivation of cereals; its conservation status was reported as good [34]. The salinity of the water samples, measured by an Aokuy refractometer (Shenzhenshi Jinshenghe Shangmao Youxiangongsi, Guangdong, China), was 50‰ *w*/*v*, and the pH measured with SRSE water test strips (Tepcom GmbH & Co., KG, Bendorf, Germany) was 7.8. The samples were transferred to 100 mL sterile plastic containers and were transported whilst being refrigerated (at 4–7 °C) to the laboratory. To maximize the diversity of isolated fungi, the following culture media were employed: 18% of glycerol agar (G18; 2.5 g peptone, 5 g dextrose, 0.5 g KH_2_PO_4_, 0.25 g MgSO_4_, 90 mL glycerol, 7.5 g agar–agar, 410 mL distilled water; [35]), potassium acetate agar (5 g potassium acetate, 1.25 g yeast extract, 0.5 g dextrose, 15 g agar–agar, 500 mL distilled water; [36]), potato dextrose agar (PDA; Laboratorios Conda S.A., Madrid, Spain; [37]) supplemented with 10% NaCl, and 2% malt extract agar (MEA; Difco Inc., Detroit, MI, USA; [38]) plus 30% glycerol. Moreover, sediment samples were activated with acetic acid following the modified protocol of Furuya and Naito [39,40]. All culture media were supplemented with 250 mg/L of L-chloramphenicol to prevent the development of bacteria. Sediment samples were vigorously shaken in the same containers they were collected in and were settled for 1 min. Once settled, water was removed by decantation and the sediment was poured onto several layers of sterile filter paper placed over plastic trays until dry [41]. Approximately, one gram of dried sediments and soil samples was sprinkled onto all the types of culture media in 90 mm Petri dishes. Different volumes of water for each of the samples (5, 15, and 30 mL) were filtered through a filter membrane of 0.45 µm diameter (Millipore SA, Molsheim, France) using a vacuum pump. Later, the filter membranes were placed onto the different culture media in 90 mm Petri dishes. Every sample was cultured by duplicate, being incubated in darkness at 15 °C and 37 °C, respectively. Plates were examined daily for up to two months by using a stereomicroscope. Each colony developed was transferred to 55 mm Petri dishes containing oatmeal agar (OA; 15 g filtered oat flakes, 7.5 g agar, 500 mL tap water; [38]) by using sterile disposable tuberculin-type needles, and these colonies were incubated at room temperature until axenic cultures of each isolate were obtained. Fungal strains suspected to be novel species or pertaining to uncommon taxa were deposited in the culture collection of the Faculty of Medicine of Reus (FMR; Reus, Tarragona Province, Spain), and the ex-type strains and the herborized specimens (as holotypes) were deposited at the Westerdijk Fungal Biodiversity Institute (CBS; Utrecht, The Netherlands).

### 2.2. Phenotypic Study

The macroscopic characterization of the colonies was performed on OA, MEA, PDA, and potato carrot agar (PCA; 10 g potato, 10 g carrot, 6.5 g agar, 500 mL distilled water) after incubation for 7–14 d at 25 °C in darkness [37,38]. The color description of the colonies was made according to Kornerup and Wanscher [42]. Cardinal growth temperatures were determined on PDA, ranging from 5 to 40 °C at 5 °C intervals, with an additional measurement at 37 °C.

The microscopic characterization of vegetative and reproductive structures was carried out by using fungal material from the colonies grown on OA under the same conditions as specified for macroscopic characterization. Measurements of at least 30 of the structures were taken from slide mountings using Shear’s medium (3 g potassium acetate, 60 mL glycerol, 90 mL ethanol 95%, 150 mL distilled water; [43]) and using an Olympus BH-2 bright field microscope (Olympus Corporation, Tokyo, Japan). Micrographs were taken employing a Zeiss Axio-Imager M1 light microscope (Zeiss, Oberkochen, Germany) with a DeltaPix Infinity × digital camera using Nomarski differential interference contrast.

### 2.3. DNA Extraction, Amplification, and Sequencing

Total genomic DNA was extracted from colonies grown on PDA for 7 to 10 days at 25 ± 1 °C in darkness following the modified protocol of Müller et al. [44] and quantified using a Nanodrop 2000 (Thermo Scientific, Madrid, Spain). For each fungal strain, we amplified the molecular marker that allowed for the most accurate preliminary identification according to the bibliography. The internal transcribed spacers (ITS) region and the D1-D2 domains of the 28S nrRNA (LSU) were amplified using the primer pairs ITS5/ITS4 [45] and LR0R/LR5 [46], respectively. Fragments of the translation elongation factor 1α (*EF-1α*) and the β-tubulin (*tub2*) genes were amplified using the primer pairs 983F/2218R and EF-728F/EF-986R [47,48] and BT2a/BT2b [49]. For our strain of interest, we amplified the following markers: ITS, LSU, *tub2*, and *EF-1α* (using the 983F/2218R set of primers). Single-band PCR products were stored at −20 °C and sequenced at Macrogen Europe (Macrogen Inc., Madrid, Spain) with the same amplification primers. Lastly, the software SeqMan v. 7.0.0 (DNAStarLasergene, Madison, WI, USA) was employed to edit and assemble the consensus sequences.

### 2.4. Phylogenetic Analysis

The sequences obtained were compared with all the sequences available at the National Center for Biotechnology Information (NCBI) database using the Basic Local Alignment Search Tool (BLAST; https://blast.ncbi.nlm.nih.gov/Blast.cgi, accessed on 11 October 2023) to obtain a preliminary molecular identification of each isolate. A maximum level of identity (MLI) of ≥98% was considered to allow for species-level identification [50]. Single and combined phylogenetic analyses of all the specific molecular markers mentioned above were initially conducted by performing a sequences alignment with the software MEGA (Molecular Evolutionary Genetics Analysis) v. 7.0. [51] using the ClustalW algorithm [52] and refining with MUSCLE [53] or/and manually, if needed. Subsequently, the phylogenetic reconstruction was made by maximum likelihood (ML) and the Bayesian Inference (BI) methods were made by two different software, RAxML-HPC2 on XSEDE v. 8.2.12 [54] software on the online CIPRES Science gateway portal [55] and MrBayes v.3.2.6 [56], respectively. The best substitution model for all the gene matrices was settled by the software from CIPRES Science gateway portal (ML) and by jModelTest v.2.1.3 following the Akaike criterion (BI) [55,57]. Regarding the ML analysis, phylogenetic support for internal branches was established by 1000 ML bootstrapped pseudoreplicates, being considered significant bootstrap support (bs) values ≥70 [58]. Regarding the BI analysis, 5 million Markov Chain Monte Carlo (MCMC) generations were used, with four runs (three heated chains and one cold chain), and samples were stored every 1000 generations. To calculate the 50% majority rule consensus tree and posterior probability values (pp), the first 25% of samples were discharged, and pp values of ≥0.95 were considered significant [59]. The resulting phylogenetic trees were plotted using FigTree v.1.3.1 (http://tree.bio.ed.ac.uk/software/figtree/, accessed on 11 October 2023). The DNA sequences and the sequence alignments generated in this study were deposited in GenBank (Table 1) and in TreeBASE (https://treebase.org, accessed on 11 October 2023), respectively. The novel taxa have been registered in MycoBank (https://www.mycobank.org/, accessed 23 March 2023 for *Dactyliodendromyces holomorphus* gen. et sp. nov.).

## 3. Results

The fungi isolated from various substrates collected in Laguna de Pito are listed in Table 2, which also includes their extremophilic character (if previously reported and/or determined during the development of our study).

Notably, among all the fungi identified, it is noteworthy that the strain FMR 20493 displayed a percentage of identity of 99.4% with *Cephalotrichum dendrocephalum* CBS 528.85 (GenBank MH873591.1; identities = 836/841; no gaps) in a BLAST search using the LSU. However, the closest hit using the ITS was *Wardomyces pulvinatus* CBS 803.69 (GenBank MH859434.1; identities = 434/447 (97.09%); two gaps), using *EF-1α* was *Wardomyces pulvinatus* CBS 112.65 and *Wardomyces humicola* CBS 369.62 (GenBank LN851102.1 and LN851097.1; for both cases: identities = 856/876 (97.7%); no gaps), and for the *tub2*, it was *Wardomyces anomalus* CBS 299.61 (GenBank LN851149.1; identities = 422/470 (89.8%); three gaps), all of them with a percentage of identity below 98%.

In the analysis involving species within the *Microascaceae*, the individual dataset for ITS, LSU, *EF-1α,* and *tub2* showed no conflicts related to the tree topologies for the 70% reciprocal bootstrap trees; thus, a multi-gene analysis was performed. The final concatenated dataset included 43 ingroup strains belonging to the genera *Acaulium*, *Cephalotrichum*, *Gamsia*, *Fairmania*, *Wardomyces* and *Wardomycopsis*, and *Microascus longirostris* CBS 196.61 and *Scopulariopsis brevicaulis* MUCL 40726 as the outgroup. The alignment encompassed a total of 3019 characters, including gaps (661 for ITS, 843 for LSU, 965 for *EF-1α,* and 550 for *tub2*), 732 of them parsimony informative (203 for ITS, 73 for LSU, 190 for *EF-1α*, and 266 for *tub2*) and 960 of them being variable sites (293 for ITS, 91 for LSU, 264 for *EF-1α*, and 312 for *tub2*). The tree obtained through the BI analysis was both congruent and similar in topology to the one obtained by ML analysis. Regarding the BI analysis, GTR + G, GTR + G + I, GTR + G + I, and HKY + G + I were selected as the models that fitted the best for ITS, LSU, *EF-1α*, and *tub2*, respectively. The support values showed slight differences between the two analysis methods, making the ML bootstrap support values lower than the BI posterior probabilities.

The phylogenetic analysis (Figure 1) revealed six fully supported clades representing the genera *Cephalotrichum* (clade I), *Gamsia* (clade VI), *Acaulium* (clade VII), *Wardomycopsis* (clade VIII), and *Fairmania* (clade IX). However, the species of the genus *Wardomyces* were placed in three fully supported independent terminal clades: clade II, comprising *Wardomyces anomalus* (the type species of the genus) and *Wardomyces inflatus*; clade III, including *Wardomyces giganteus* (basionym *Microascus giganteus*) and *Wardomyces ovalis*; and clade IV, composed of *Wardomyces humicola* and *Wardomyces pulvinatus*. Furthermore, our strain CBS 149968 was placed as an independent terminal clade itself (clade V). Therefore, clades III, IV, and V represent three novel genera, also supported by phenotypic features.

### Taxonomy

*Microascaceae* Luttrell et Malloch, Mycologia 62:734 (1970). Mycobank MB 81001.

*Ascomata* globose, pyriform or irregular in shape, dark brown to black, hairy, rarely bare, arising from coiled ascocarp initials, with or without an ostiole; *asci* arising singly or in chains on the ascogenous hyphae, without croziers, ovoid to globose, soon evanescent; *ascospores* reddish brown to coppery-colored, one-celled, with a germ pore at one or both ends, dextrinoid when young, smooth- and thin-walled.

*Dactyliodendromyces* Barnés-Guirado, Cano & Stchigel, gen. nov. MycoBank MB 848097.

*Etymology*. From Greek *δακτύλιος*- (daktýlios), anything ring-shaped, -*δένδρον*- (déndron), tree, and -*μύκητας* (mýkitas), fungus, because the fungus produces tree-like conidiophores bearing anellidic conidiogenous cells.

*Description: Hyphae* hyaline to subhyaline, septate, smooth-walled to asperulate, thin-walled, branched, sometimes aggregated and frequently anastomosing. *Asexual morph*—*Conidiophores* macronematous, penicillate, branching up to three times, subhyaline to pale brown or pale olivaceous. *Conidiogenous cells* annellidic, mono- or polyblastic, terminal, discrete, flask-shaped, ventricose, with a short terminal neck. *Conidia* solitary or disposed in short basipetal chains on the conidiogenous cell, one-celled, pale brown to brown, smooth- and thick-walled, ovoid to lenticular, flattened at the base, without germ slits or pores, secession schizolytic. *Sexual morph*—*Ascomata* erumpent, dark greyish brown when mature, ostiolate, setose, globose to subglobose, neck short, cylindrical; setae dark brown, septate; peridium superficially areolate when young, becoming carbonaceous with the age, of *textura angularis*, formed by an outer wall of dark brown polygonal cells, and an inner wall of hyaline to pale brown polygonal cells. *Asci* 8-spored, broad ellipsoidal to ovoid, soon evanescent. *Ascospores* one-celled, apricot to pale orange, hearth- or kidney-shaped, small, with a terminal germ pore.

*Type species*: *Dactyliodendromyces holomorphus* Barnés-Guirado, Cano & Stchigel, sp. nov. MycoBank MB 848098.

*Dactyliodendromyces holomorphus* Barnés-Guirado, Cano & Stchigel, sp. nov. MycoBank MB 848098 (Figure 2).

*Etymology*. From Greek *όλος—*(ólos), the whole, and -*μορφή* (morfí), form, because the fungus produces both sexual and asexual morphs.

*Description: Hyphae* hyaline to subhyaline, septate, smooth-walled to asperulate, thin-walled, branched, sometimes aggregated and frequently anastomosing, 1.0–3.0 µm wide. *Asexual morph*—*Conidiophores* macronematous, penicillate, 1–3-branched, smooth- and thick-walled, subhyaline to pale brown or pale olivaceous, 20–55 µm long, 2.0–3.0 µm wide at the base; *branches* smooth- and thick-walled, 2–3.5 µm wide and 4–3.5 µm long, primary branches bearing 2 to 4 secondary branches, secondary branches bearing 1 to 2 tertiary branches, and the terminal branches bearing 1 to 5 conidiogenous cells. *Conidiogenous cells* annellidic, mono- or polyblastic, terminal, discrete, smooth- and thick-walled, hyaline to pale brown or pale olivaceous, flask-shaped, ventricose with a short terminal neck, 3.5–6 × 1–3 µm, bearing a terminal conidium or conidia disposed in short chains. *Conidia* one-celled, pale brown to brown, smooth- and thick-walled, ovoid to lenticular, 5–8 × 2–3 µm, rounded at the apex and flattened at the base, without germ slits or pores, secession schizolytic. *Sexual morph*—*Ascomata* erumpent, usually formed at the periphery of the colony, hyaline to pale brown when young, becoming dark greyish brown to very dark brown when mature, areolate when young, ostiolate, tomentose, setose, piriform, body without the neck globose to subglobose, 211–327 × 220–336 µm; *neck* short, up to 55 µm, cylindrical, 36–41 × 26.5–35 µm; *setae* smoky olivaceous brown to dark brown, septate, smooth- and thick-walled, needle-shaped but with a sinuous wall, 16.5–450 × 1–5 µm, mostly tapering and paler towards the top; *peridium* at first areolate, of *textura angularis*, becoming carbonaceous with the age, formed by an outer wall of dark brown polygonal cells, and an inner wall of hyaline to pale brown polygonal cells; not easily breakable under external pressure. *Asci* 8-spored, broadly ellipsoidal to ovoid, 10 × 6–8 µm, soon evanescent, catenated when young. *Ascospores* one-celled, apricot to pale orange, smooth- and thin-walled, heart-shaped to kidney-shaped but flattened at one side, 3–4 × 2.5–3 × 2 µm, with an inconspicuous germ pore at one of the extremes.

*Culture characteristics* (after 14 d at 25 °C)—Colonies on PDA reaching 11 mm diam., convex, smooth texture, cerebriform, white (1A1), undulate, sporulation absent; reverse white, (1A1), and olive brown (4E6) at centre, white (1A1) towards periphery, soluble pigment absent. Colonies on PCA reaching 18 mm diam., slightly raised at centre, flattened at the edges, granulose, smooth, grey (30F1) at centre, white (1A1) at the edges, filamentous margins, sporulation (conidiophores) moderate; reverse white (1A1), soluble pigment absent. Colonies on OA reaching 24 mm diam., flattened, granulose, smooth, grey (30F1) at centre, white (1A1) towards periphery, filamentous margins, sporulation moderate to abundant (conidiophores); reverse olive (3F5) at centre, white (1A1) towards periphery, soluble pigment absent. Non-halophilic nor highly halotolerant (does not grow above 10% *w*/*v* NaCl). *Cardinal temperatures of growth*: minimum 5 °C, optimum 20 °C, maximum 30 °C.

*Specimen*: CBS 149968. Spain, Aragon community, Zaragoza province, Laguna de Pito (41°24′44.2″ N 0°09′02.2″ W), isolated from lagoon sediment, 17 January 2022, collected by María Barnés Guirado, Alan Omar Granados Casas, Alberto Miguel Stchigel Glikman and José F. Cano-Lira, isolated by María Barnés Guirado, holotype CBS H-25252.

*Diagnosis*: The sexual morph of the genus *Dactyliodendromyces* resembles those species of the *Microascaceae* producing heart-shaped ascospores: *Acaulium albonigrescens*, *Fairmania singularis*, and *Wardomyces giganteus*, as well as several species of *Microascus* and *Scopulariopsis*. Nevertheless, *Dactyliodendromyces* produces ostiolate ascomata with a superficially areolate peridium and true setae, features not seen in the other taxa. On the other hand, the asexual morph of genus *Dactyliodendromyces* differs from *Wardomyces*, and the newly proposed genera *Parawardomyces* and *Pseudowardomyces*, in having annellidic conidiogenous cells, which are holoblastic in all of them. The genus *Gamsia* differs from *Dactyliodendromyces* by the formation of hyaline, mostly undifferentiated and unbranched conidiophores bearing polyblastic and anellidic conidiogenous cells, which are dematiaceous, well-developed, penicillated, and bear exclusively annellidic conidiogenous cells in *Dactyliodendromyces*. Consequently, the asexual morph of *Dactyliodendromyces* is morphologically more similar to *Acaulium, Cephalosporium*, *Fairmania*, and *Wardomycopsis*. However, *Dactyliodendromyces* is easily discriminated from *Fairmania* and *Wardomycopsis* because these genera produce conidia with longitudinal striations or germ slits (not seen in *Dactyliodendromyces*), and from *Cephalosporium*, because it produces conidiophores grouped in synnemata (which are absent in *Dactyliodendromyces*), and also lacks a sexual morph (present in *Dactyliodendromyces*). *Acaulium*, in comparison to the *Dactyliodendromyces*, produces more simple hyaline conidiophores, which are dematiaceous and penicillate in the new genus.

*Pseudowardomyces* Barnés-Guirado, Stchigel & Cano, gen. nov. Mycobank MB 851965.

*Etymology*: From Greek *ψευδο*- (psevdo), false, because of its morphological resemblance to the genus *Wardomyces*.

*Description: Hyphae* hyaline, septate, smooth- and thin-walled, branched, sometimes aggregated and frequently anastomosing. *Conidiophores* hyaline, macronematous, mostly branched, bi- to terverticillate, with a stipe of short to medium length. *Conidiogenous cells* holoblastic, terminal or subterminal, globose to barrel-shaped, producing one to three conidia per cell. *Conidia* two-celled, smooth- and thick-walled, navicular, slightly constricted at the septum, upper cell ovoid with a truncate base, subacute at the apex, dark brown, with a longitudinal pale-colored germ slit, basal cell smaller and hyaline, irregularly barrel-shaped to campaniform, secession rhexolitic. *Sexual morph* not observed.

*Type species*: *Pseudowardomyces humicola* (Hennebert & G.L. Barron) Barnés-Guirado, Stchigel & Cano, comb. nov. MycoBank MB851997. *Wardomyces humicola* Hennebert & G.L. Barron, Can. J. Bot. 40: 1209 (1962). [Basionym].

*Other species*: *Pseudowardomyces pulvinatus* (Marchal) Barnés-Guirado, Stchigel & Cano, comb. nov. MycoBank MB851998. *Echinobotryum pulvinatum* Marchal, Bull. Soc. R. Bot. Belg. 34(no. 1): 139 (1895). [Basionym].

*Diagnosis*: The genus *Pseudowardomyces* is morphologically similar to the genus *Wardomyces* but differs in the production of more complex conidiophores. On the other hand, *Pseudowardomyces* differs morphologically from *Parawardomyces* by the production of two-celled conidia (unicellular in *Parawardomyces*).

*Parawardomyces* Barnés-Guirado, Stchigel & Cano gen. nov. MycoBank MB851964.

*Etymology*: From Greek *παρα*- (para) due to its morphological resemblance to the genus *Wardomyces*.

*Description*: *Hyphae* hyaline, septate, smooth- and thin-walled, branched. Asexual morph *Conidiophores* micronematous to semi-macronematous, monoverticillate, short-stipitate, hyaline. *Conidiogenous cells* holoblastic, terminal, short, cylindrical to barrel-shaped, producing usually one to three, sometimes more conidia per cell. *Conidia* one-celled, hyaline to pale brown, solitary, ellipsoid to cylindrical with a rounded apex and a truncate base, smooth-walled, with a longitudinal pale-colored germ slit, secession schizolytic. Scopulariopsis-like synanamorph present. *Conidiophores* micronematous to macronematous, biverticillate, stipitate, penicillate. *Conidiogenous cells* annellidic, flask-shaped, solitary, in whorls of three to five on the vegetative hyphae, or on verticillate metulae. *Conidia* one-celled, smooth- and thick-walled, pyriform to ovoid, basally truncate, in basipetal chains. *Sexual morph*—*Ascomata* dark brown, ostiolate, setose, subglobose to globose; neck long, cylindrical, setose; *peridial wall* of *textura angularis*. *Asci* subglobose to globose, thin-walled, non-stipitate, eight-spored, soon evanescent. *Ascospores* one-celled, subhyaline to pale orange, smooth-walled, kidney-shaped, flattened laterally, with a germ pore at each end, germinating by means of germ tubes through one or both pores.

*Type species*: *Parawardomyces ovalis* (W. Gams) Barnés-Guirado, Stchigel & Cano, comb. nov. MycoBank MB 851999. *Wardomyces ovalis* W. Gams, Trans. Br. mycol. Soc. 51(5): 798 (1968). [Basionym].

*Other species*: *Parawardomyces giganteus* (Malloch) Barnés-Guirado, Stchigel & Cano, comb. nov. MycoBank MB 852000. *Microascus giganteus* Malloch, Mycologia 62(4): 731 (1970). [Basionym].

*Diagnosis*: *Parawardomyces* differs from *Pseudowardomyces* and *Wardomyces* by the production of unicellular conidia (bicellular in the latter genera), and by a scopulariopsis-like synanamorph (absent in the other two genera). In addition, one of their species (*Parawardomyces gigantea*) produces a microascus-like sexual morph, which is absent in both *Pseudowardomyces* and *Wardomyces*.

Due to the reassignment of some species that belonged to the genus *Wardomyces* into other genera, we have amended it as follows.

*Wardomyces* F.T. Brooks & Hansf. Trans. Br. mycol. Soc. 8(3): 137 (1923). MycoBank MB10433.

*Description: Hyphae* hyaline, branched, sometimes aggregated and septate. *Conidiophores* semi-macronematous, mononematous, mostly biverticillate, sometimes terverticillate, short-stipitate, straight, hyaline, smooth, and branched. *Conidiogenous cells* polyblastic, determinate, ampulliform, doliiform, or irregularly shaped. *Conidia* solitary, ovoid, sometimes pointed at the apical end, ellipsoidal to slightly cylindrical, truncated at the base, brown or blackish brown, smooth, with a longitudinal germ slit, aseptate, secession schizolytic. *Sexual morph* not observed.

*Type species: Wardomyces anomalus* F.T. Brooks & Hansf. [as ‘anomala’], Trans. Br. mycol. Soc. 8(3): 137 (1923). MycoBank MB256937.

Other species: *Wardomyces inflatus* (Marchal) Hennebert. MycoBank MB341001. *Trichosporum inflatum* Marchal, Champ. copr. Belg. 7: 142 (1896). [Basionym].

*Diagnosis*: *Wardomyces* differs from *Parawardomyces* and *Pseudowardomyces* by presenting semi-macronematous, short-stipitate, mostly biverticillate conidiophores. Moreover, it differs from *Parawardomyces* because does not present the scopulariopsis-like synanamorph and lacks a sexual morph, and from *Pseudowardomyces* in the conidial shape, the absence of a septum and its schizolytic secession.

## 4. Discussion

Although some fungi found during the course of this study, such as *Alternaria alternata*, *Aspergillus amstelodami*, and *Aspergillus versicolor*, have been previously reported in hypersaline lagoons and lakes [75], there are no reports of these identified species in endorheic lagoons in Spain. Notably, the extremophilic nature of several species belonging to globally distributed genera recovered in this study, such as *Aspergillus*, *Cladosporium*, and *Penicillium*, has been documented in earlier studies [76,77,78,79]. Additionally, most of the identified species have been previously reported as extremophilic or extremotolerant. For instance, *Aspergillus amstelodami*, *Aspergillus intermedius*, and *Penicillium egyptiacum* exhibit osmophilic, xerophilic, or xerotolerant behavior [64,72], while *Parachaetomium truncatulum*, *Aspergillus calidoustus*, and *Chaetomium grande* display thermotolerance [64,70].

Particularly interesting are the findings of *Chaetomium grande* and *Parachaetomium truncatulum*, which represent two new reports for Spain and Europe, also being the first time that this species has been isolated from lake sediments [70,80,81,82]. Furthermore, *Acrostalagmus luteoalbus* can thrive in soils with high pH values (alkali-tolerant), and *Aspergillus versicolor* and *Stachybotrys chartarum* exhibit halotolerance [60,68,74]. Some species have not been previously reported as extremophilic or extremotolerant, yet our study reveals their ability to grow in up to 10% *w*/*v* NaCl, such as *Cladosporium europaeum* and *Malbranchea zuffiana*. Surprisingly, we recovered *Cephalotrichiella penicillata*, *Fusarium culmorum*, and *Ovatospora amygdalispora*, taxa that do not exhibit any extremophilic/extremotolerant characteristics and, consequently, should be considered as non-specialized.

Based on both phylogenetic analysis and phenotypic features, we introduced the new monotypic genus *Dactyliodendromyces*, isolated from a sediment sample from *Laguna de Pito*. This fungus does not exhibit a strong halotolerant behavior and belongs to the *Microascaceae* family. While members of *Microascaceae* have a global distribution, only a few have been initially discovered in Spain, such as *Wardomycopsis litoralis* and *Pseudoscopulariopsis schumacheri* [20,83]. Although most of the members of *Microascaceae* are not usually isolated from extreme environments [29,84,85,86], some species of the genera *Microascus* and *Scopulariopsis* have been isolated from halophyte plants and salt marshes, respectively [87,88]. Moreover, only two species of this family were first isolated from salty habitats: *Wardomyces pulvinatus* and *Wardomycopsis litoralis* [83,89]. *Dactyliodendromyces holomorphus* differs from its closely related genera *Gamsia*, *Parawardomyces*, and *Pseudowardomyces* by the production of the holomorph in vitro. The only exception is *Parawardomyces giganteus*, which also produces a sexual morph, yet they do not morphologically resemble each other [19]. The sexual morph of *D. holomorphus* consists of short-naked ostiolate ascomata with true setae and a tomentose, carbonaceous peridium, with these features being uncommon among the *Microascaeae* [24,85,90,91].

Despite previous attempts to separate the genus *Wardomyces* into different genera, nowadays, it is still considered a paraphyletic genus [92]. However, based on our phylogenetic analysis using the ITS-LSU-*EF-1α*-*tub2* markers, *Wardomyces* could be segregated into three different genera. Furthermore, based on the morphological features, *Wardomyces* produces complex conidiophores, bi- to terverticillate, and one-celled conidia, whereas *Parawardomyces* is characterized by the production of monoverticillate conidiophores, and a scopulariopsis-like and a mammaria-like synanamorph, whereas *Pseudowardomyces* is distinguished by the production of bi-celled conidia.

To date, there is limited information on fungi isolated from endorheic lagoons in Europe, including Spain. Therefore, this study makes a significant contribution to the understanding of mycobiota in such environments by documenting the discovery of a new genus of the order Microascales and several rare taxa, particularly from the *Chaetomiaceae* (order Sordariales) family. It is noteworthy that no extremely halophilic fungi were identified.

## Figures and Tables

**Figure 1 jof-10-00236-f001:**
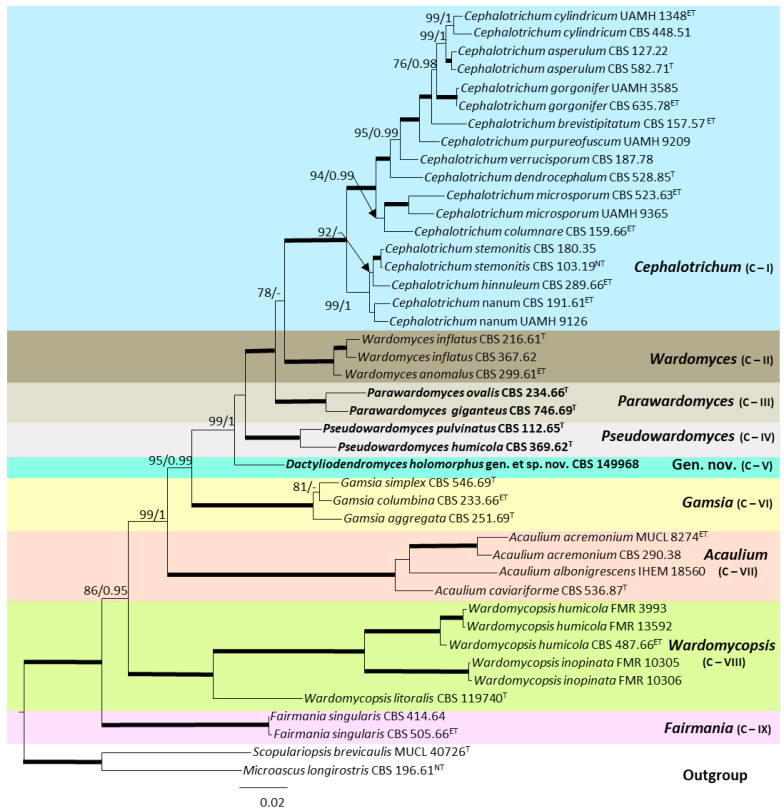
Maximum likelihood phylogenetic tree obtained by combining ITS, LSU, *EF-1α,* and *tub2* sequences from 43 representative taxa of the *Microascaceae*. RAxML bootstrap support (BS) values and Bayesian posterior probabilities (PP) greater than 70% and 0.95, respectively, are shown above the branches. Fully supported branches (100% BS/1 PP) are indicated as broad lines. Novel genera are indicated in bold. The tree was rooted to *Microascus longirostris* CBS 196.61 and *Scopulariopsis brevicaulis* MUCL 40726. ^T^ = Ex-type; ^ET^ = Ex-epitype; ^NT^ = Ex-neotype.

**Figure 2 jof-10-00236-f002:**
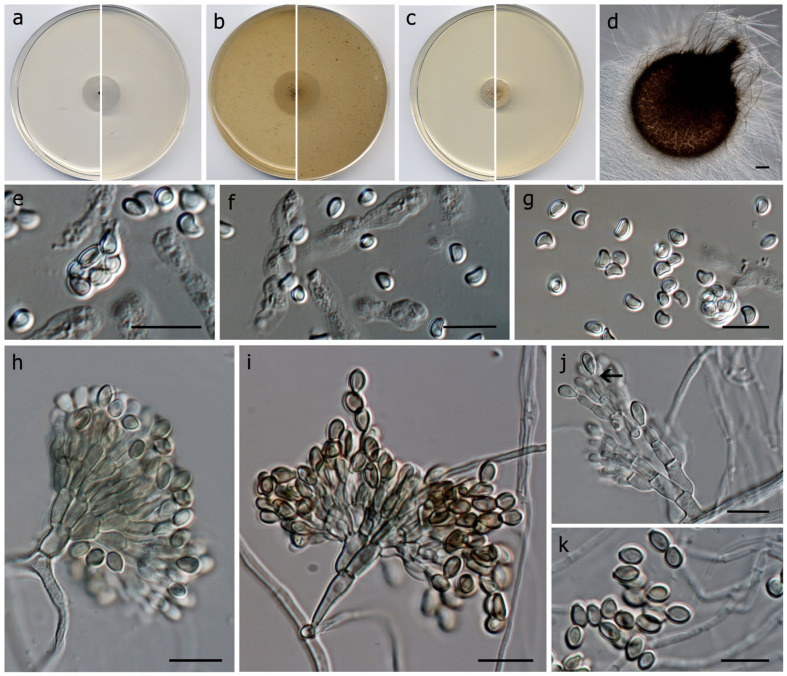
*Dactyliodendromyces holomorphus* CBS 149968. Colonies on PCA (**a**), OA (**b**), and PDA (**c**) after two weeks at 25 + 1 °C (surface, left; reverse, right); (**d**) young ascoma; (**e**) asci; (**f**) catenated asci; (**g**) ascospores; (**h**,**i**) Conidiophores; (**j**) Conidiogenous cells, arrow points out annelid’s rings; (**k**) conidia. Scale bars: (**e**–**k**) = 10 µm; (**d**) = 25 µm.

**Table 1 jof-10-00236-t001:** Fungi and nucleotide sequences of the molecular markers used to build the phylogenetic trees.

Taxon	Strain Number	Source	Origin	Sequence Accession Number			
				LSU	ITS	*EF*-1*α*	*tub*2
*Acaulium acremonium*	CBS 290.38	Skin of a horse	København, Denmark	LN851001	LM652456	HG380362	LN851108
*Acaulium acremonium*	MUCL 8274 ^ET^	Wheat field soil	Schleswig-Holstein, Germany	LN851002	LM652457	LN851056	LN851109
*Acaulium albonigrescens*	IHEM 18560 ^ET^	Litter treated with urea	Nemuro-shi, Japan	LN851004	LM652389	LN851058	LN851111
*Acaulium caviariformis*	CBS 536.87 ^ET^	Decaying meat	Flemalle, Belgium	LN851005	LM652392	LN851059	LN851112
*Cephalotrichum asperulus*	CBS 127.22	Seed	Wageningen, The Netherlands	LN851006	LN850959	LN851060	LN851113
*Cephalotrichum asperulus*	CBS 582.71 ^IT^	Soil	Buenos Aires, Argentina	LN851007	LN850960	LN851061	LN851114
*Cephalotrichum brevistipitatum*	CBS 157.57 ^T^	Tuber	Wageningen, The Netherlands	LN851031	LN850984	LN851084	LN851138
*Cephalotrichum columnare*	CBS 159.66 ^T^	Dung of hare	Johannesburg, South Africa	LN851010	LN850963	LN851064	LN851117
*Cephalotrichum cylindricum*	CBS 448.51	Timber	Bekker, South Africa	LN851011	LN850964	LN851065	LN851118
*Cephalotrichum cylindricum*	UAMH 1348 ^ET^	Seed of sorghum	KS, USA	LN851012	LN850965	LN851066	LN851119
*Cephalotrichum dendrocephalum*	CBS 528.85 ^IT^	Cultivated soil	Basrah, Iraq	LN851013	LN850966	LN851067	LN851120
*Cephalotrichum gorgonifer*	CBS 635.78 ^ET^	Hair	The Netherlands	LN851024	LN850977	LN851077	LN851131
*Cephalotrichum gorgonifer*	UAMH 3585	Mushroom compost	Spruce Grove, AB, Canada	LN851025	LN850978	LN851078	LN851132
*Cephalotrichum hinnuleum*	CBS 289.66 ^T^	Dung of deer	Tasmania, Australia	LN851032	LN850985	LN851085	LN851139
*Cephalotrichum microsporum*	CBS 523.63 ^ET^	Wheat field soil	Schleswig-Holstein, Germany	LN851014	LN850967	LN851068	LN851121
*Cephalotrichum microsporum*	UAMH 9365	Indoor air	Peace River, AB, Canada	LN851015	LN850968	LN851069	LN851122
*Cephalotrichum nanum*	CBS 191.61 ^ET^	Dung of deer	Richmond Park, SRY, ENG, UK	LN851016	LN850969	LN851070	LN851123
*Cephalotrichum nanum*	UAMH 9126	Dung of bison	Elk Island National Park, AB, Canada	LN851017	LN850970	LN851071	LN851124
*Cephalotrichum purpureofuscum*	UAMH 9209	Indoor air	Pemberton, BC, Canada	LN851018	LN850971	LN851072	LN851125
*Cephalotrichum stemonitis*	CBS 103.19 ^NT^	Seed	Wageningen, The Netherlands	LN850952	LN850951	LN850953	LN850954
*Cephalotrichum stemonitis*	CBS 180.35	Unknown	Unknown	LN851019	LN850972	LN851073	LN851126
*Cephalotrichum verrucisporum*	CBS 187.78	Dune soil	Katijk, The Netherlands	LN851033	LN850986	LN851086	LN851140
** *Dactyliodendromyces holomorphus* **	CBS 149968	Lagoon sediment	Zaragoza, Spain	OR141719	OR141718	OR142400	OR142401
*Fairmania singularis*	CBS 414.64	Laboratory contaminant	Tokyo, Japan	LN851035	LM652442	LN851088	LN851142
*Fairmania singularis*	CBS 505.66 ^ET^	Barrel bottom	Kittery Point, ME, USA	LN851036	LN850988	LN851089	LN851143
*Gamsia aggregata*	CBS 251.69 ^IT^	Dung of carnivore	Wycamp Lake, MI, USA	LN851037	LM652378	LN851090	LN851144
*Gamsia columbina*	CBS 546.69 ^T^	Milled *Oryza sativa*	Osaka, Japan	LN851041	LM652379	LN851094	LN851148
*Gamsia columbina*	CBS 233.66 ^ET^	Sandy soil	Giessen, Germany	LN851039	LN850990	LN851092	LN851146
** *Parawardomyces ovalis* **	CBS 234.66 ^T^	Wheat field soil	Schleswig-Holstein, Germany	LN851050	LN850996	LN851101	LN851155
** *Parawardomyces giganteus* **	CBS 746.69 ^T^	Insect frass in a dead log	Coldwater, ON, Canada	LN851045	LM652411	LN851096	LN851150
** *Pseudowardomyces pulvinatus* **	CBS 112.65 ^T^	Salt marsh	CHS, ENG, UK	LN851051	LN850997	LN851102	LN851156
** *Pseudowardomyces humicola* **	CBS 369.62 ^IT^	Soil in tropical greenhouse	Guelph, ON, Canada	LN851046	LN850993	LN851097	LN851151
*Scopulariopsis brevicaulis*	MUCL 40726 ^T^	Indoor air	Scandia, AB, Canada	LN851042	LM652465	HG380363	LM652672
*Microascus longirostris*	CBS 196.61 ^NT^	Wasp’s nest	Kittery Point, ME, USA	LN851043	LM652421	LM652566	LM652634
*Wardomyces anomalus*	CBS 299.61 ^ET^	Air cell of egg	Ottawa, ON, Canada	LN851044	LN850992	LN851095	LN851149
*Wardomyces inflatus*	CBS 216.61 ^IT^	Wood, *Acer* sp.	Sainte-Cécile-de-Masham, QC, Canada	LN851047	LM652496	LN851098	LN851152
*Wardomyces inflatus*	CBS 367.62 ^NT^	Greenhouse soil	Heverlee, Belgium	LN851048	LN850994	LN851099	LN851153
*Wardomycopsis humicola*	CBS 487.66 ^IT^	Soil	Guelph, ON, Canada	LM652554	LM652497	LN851103	LN851157
*Wardomycopsis humicola*	FMR 3993	Sediment of Ter River	Girona, Spain	LN851052	LN850998	LN851104	LN851158
*Wardomycopsis humicola*	FMR 13592	Soil	Reus, Spain	LN851053	LN850999	LN851105	LN851159
*Wardomycopsis inopinata*	FMR 10305	Soil	Bagan, Myanmar	LN851054	LM652498	LN851106	LN851160
*Wardomycopsis inopinata*	FMR 10306	Soil	Bagan, Myanmar	LN850956	LN850955	LN850957	LN850958
*Wardomycopsis litoralis*	CBS 119740 ^T^	Beach soil	Castellon, Spain	LN851055	LN851000	LN851107	LN851161

CBS, CBS-KNAW Westerdijk Fungal Biodiversity Institute (Utrecht, the Netherlands). FMR, Facultat de Medicina Reus (URV—Reus—Spain). IHEM, BCCM/IHEM Belgian Fungi Collection: Human and Animal Health. MUCL, BCCM/MUCL Belgian Agro-food and Environmental Fungal Collection. UAMH, University of Alberta Mold Herbarium and Culture Collection (Edmonton, Canada). New taxa are in **bold**. ^ET^, ex-epitype strain. ^IT^, ex-isotype strain. ^NT^, ex-neotype strain. ^T^, ex-type strain.

**Table 2 jof-10-00236-t002:** Fungal taxa recovered from Laguna de Pito and their extremophilic properties.

Taxon	Strain Nr ^1^	Identity Percentage (%)	GenBank Accession Nr ^2^	Markers Used	Source	Extremophilic Features Reported	References
*Acrostalagmus luteoalbus*	19813 *	99.69	KP050692	ITS	water	Alkali-tolerant	[60]
*Actinomucor elegans*	19823 *	99.79	AY243954	ITS	water	Thermotolerance (strain-dependent)	[61]
*Alternaria alternata*	20034 *	100	KP124364	ITS	sediment	Halotolerant; alkali-tolerant	[62]
*Alternaria chlamydospora*	20037 *	100	MG020753	ITS	sediment	Acidophilic; alkali-tolerant; psychrotolerant; xerotolerant	[63]
*Aspergillus amstelodami*	20038 *	99.22	MT820427	*tub2*	water	Xerophilic; thermotolerant	[64,65]
*Aspergillus calidoustus*	19423 *, 19820	99.7	LT798990	*tub2*	water	Thermotolerant	[64]
*Aspergillus intermedius*	19821 *	100	LT671082	*tub2*	water	Osmophilic; thermotolerant; xerophilic	[66]
*Aspergillus montevidensis*	20492 *	100	KF499570	*tub2*	sediment	Halotolerant	[67]
*Aspergillus versicolor*	19427 *, 20659	100	ON807694	*tub2*	sediment	Halotolerant	[68]
*Cephalotrichiella penicillata*	20498 *	100	NR_153893	ITS	sediment	Not reported	[69]
*Chaetomium grande*	20036 *	99.30	KT214731	*tub2*	sediment	Thermotolerant	[70]
*Cladosporium europaeum*	19425 *, 20499	100	HM148294	*EF-1α*	water	Halotolerant	Our study
*Dactyliodendromyces holomorphus*	20493 *	On text		ITS, LSU, *EF-1α*, *tub2*	sediment	Not reported	Our study
*Epicoccum italicum*	20044 *	100	MN983956	*tub2*	sediment	Psychrotolerant; halotolerant	[71]
*Fusarium culmorum*	20248 *	99.85	KT008433	*EF-1α*	sediment	Not reported	
*Malbranchea zuffiana*	20033 *	98.90	MH869293	ITS	sediment	Halotolerant	Our study
*Ovatospora amygdalispora*	20322 *	99.41	MZ343030	*tub2*	sediment	Not reported	
*Parachaetomium truncatulum*	20041 *, 20495	99.77	HM365298	*tub2*	sediment	Thermotolerant	[70]
*Penicillium egyptiacum*	20328 *, 20324, 20323, 20331, 20337	100	JX996851	*tub2*	sediment	Psychrotolerant; non-thermotolerant; xerotolerant	[72]
*Sordaria fimicola*	19587 *	99.61	MH860820	ITS	sediment	Striking stimulation of ascospore germination by acetate	[73]
*Stachybotrys chartarum*	19808 *	100	KU846678	ITS	sediment	Halotolerant	[74]

* Strain sequenced. ^1^ FMR, Facultat de Medicina Reus (URV—Reus—Spain). ^2^ Nucleotide sequence for which the best match has been recorded.

## Data Availability

Data are contained within the article.

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
