# Peer review of "A New Genus of the Microascaceae (Ascomycota) Family from a Hypersaline Lagoon in Spain and the Delimitation of the Genus Wardomyces"

_jof, 2024, doi:10.3390/jof10040236_

Round 1

Reviewer 1 Report

This paper is well written. My suggestions for this article are minor revisions, some details need to be worked out further.

1. The title should be modified.

2. More information about taxonomy of Microascaceae should be added in the introduction.

3. More details should be modified and added in the Table 2. For example, accession number of reference sequence should be added. In the "Percentage (%) of identity" column, "99,69%" should be "99.69"? Please check carefully.

4. In the figure 2, please changing the order of microscopic photos so that the sexual form is at the front and the asexual form at the back, i.e. "i, j, k" in this version should be in front of "e, f, j, h".

Author Response

Dear reviewer,

Thank you so much for your constructive evaluation of our manuscript.

According to your comments:

- we changed the title of our paper to “A new genus of the family Microascaceae (Ascomycota) from a hypersaline lagoon in Spain, and the delimitation of the genus Wardomyces”.

- we introduced more taxonomic information about the family Microascaceae (page 2, lines 73 to 95).

- we modified Table 2, including the GenBank accession numbers of the nearest taxon in the same row than “Percentage (%) of identity”, to not increase the size of the table.

And,

- we changed the order of the pictures so that the sexual structures come first and the asexual ones second.

All changes in the new version of the manuscript are highlighted in yellow background.

Best regards,

The authors.

Reviewer 2 Report

The work by María Barnés-Guirado et al. describes the identification and characterization of the fungal diversity inhabiting the Laguna de Pito in Las Saladas de Sástago-Bujaraloz, Spain. This was a worth work to be done, in which the authors suggested the phylogenetic placement of isolated fungi using an massive morphological depiction and the most modern molecular methods, even suggesting a new fungi species.

Their extensive work in identifying local fungi should be recognized, and used as an example for other latitudes.

Minor corrections:

Line 136. … grown on PDA for 7 to 10 days at 25 ± 1 ºC in darkness following… Correct the ° symbol.

Caption of Fig. 2 The description of (c) is missing. Probably is PDA, but it is not written in the text.

Author Response

Dear reviewer,

Thanks a lot for the promising review of our article.

The corrections were made according to your suggestions, and are highlighted in yellow background in the new version of the manuscript.

Best regards,

The authors

Reviewer 3 Report

this study is really important and interesting. authors designed the study very well and provided the information. There are a few things that can be improved. I have added those in the manus. 

You can provide the number of genera in the family and cite some important studies. 

Author Response

Dear reviewer,

Thank you very much for your encouraging evaluation of our article.

All your corrections and suggestions were accepted by us, and are included (highlighted in yellow background) in the new version of the manuscript.

Best regards,

The authors